# A Pilot Study to Examine the Effects of a Workplace Cyberbullying Cognitive Rehearsal Mobile Learning Program for Head Nurses: A Quasi-Experimental Study

**DOI:** 10.3390/healthcare11142041

**Published:** 2023-07-17

**Authors:** Mijeong Park, Ok Yeon Cho, Jeong Sil Choi

**Affiliations:** 1Department of Nursing, Hoseo University, 20, Hoseo-ro 79, Baebang-eup, Asan-si 31499, Republic of Korea; mijeong@hoseo.edu; 2Department of Nursing, Gachon University Gil Medical Center, Incheon 21565, Republic of Korea; oycho@gilhospital.com; 3College of Nursing, Gachon University, 191, Hambakmoe-ro, Yeonsu-dong, Yeonsu-gu, Incheon 21936, Republic of Korea

**Keywords:** nurses, mobile learning, cyberbullying, workplace cyberbullying

## Abstract

This study aimed to (1) develop a version of the cognitive rehearsal program that is suitable for cyberbullying and (2) apply the m-learning method to evaluate its effectiveness among head nurses. This study was conducted in July 2021 and comprised 69 South Korean university hospital head nurses. It was evaluated using a nonequivalent control group pretest-posttest and a quasi-experimental design. The program was developed using the Analysis, Design, Development, Implementation, and Evaluation (ADDIE) step process, consisting of 10 scenarios. Variables with proven reliability were used in the program effect measurement. The differences between the experimental and control groups were examined using an independent *t*-test (perception) or the Mann–Whitney U test (symptom experience, knowledge, and turnover intention). The program positively affected and improved head nurses’ knowledge and perception of workplace cyberbullying; however, it elevated their symptom experience and had no immediate impact on turnover intention. The developed program could be applied as a valuable educational strategy in the nursing field. Head nurses act as intermediaries between individuals and the organization. Therefore, they must respond with in-depth knowledge and perceptions of cyberbullying to fulfill their responsibilities of identifying, mediating, and managing cyberbullying among hospital team members.

## 1. Introduction

Bullying in nursing organizations is more common than in other establishments [1,2]. While electronic communication using computers or smartphones can serve as a fast and effective means of work in nursing [1], it can also be used to instigate workplace cyberbullying. Cyberbullying refers to a situation in which individuals are repeatedly subjected to perceived negative acts conducted through technology at their workplace [3]. While cyberbullying has been extensively studied in the literature on childhood adolescence and emerging adulthood [4], only recently has systematic research begun to focus on cyberbullying in the workplace [5,6,7].

Based on a literature review and meta-analysis, a number of theories serve as the theoretical foundation for the causes and effects of workplace cyberbullying [8,9,10,11]. The Affective Events Theory (AET) describes how the experience of emotions like anger or fear can cause affective work events to result in different work attitudes and affective-driven behaviors [8]. Initially intended as a theory of work satisfaction, AET integrated prior knowledge of emotions to provide future directions for the research on emotions in the organizational context.

Cyberbullying, which occurs in cyberspace, is more serious than traditional bullying, as victims can be continuously tormented regardless of their physical location [5]. Cyberbullying can be identified based on a socio-ecological model [2,10], which has previously also been used to examine traditional workplace bullying in nursing contexts [10]. Although not a complete list of contributing factors, the socio-ecological model may serve as a starting point for enhancing our understanding of how workplace cyberbullying occurs. Additionally, its contributing factors may serve as a springboard for management, intervention, and prevention initiatives against workplace cyberbullying. Based on the socio-ecological model, workplace cyberbullying occurs in three levels: individual (micro), organizational (meso), and industry and national (macro); its outcomes have several influences on each level [2]. At the individual level, workplace cyberbullying negatively impacts the victims’ psychological and physical well-being [12]. At the organizational level, according to the AET theory, it has an effect on turnover intention to leave an organization and job satisfaction [8].

Thus far, interventions aimed at preventing workplace bullying include the cognitive rehearsal program by Griffin [13], in which new nurses rehearsed coping methods for bullying scenarios. This behavioral technique is often employed in cognitive behavioral therapy, which encompasses practicing specific scenarios for appropriate interaction, or positive coping processes [13]. It increases nurses’ knowledge, confidence, and awareness of workplace bullying [14], and decreases the turnover rate [13] or turnover intention [15]. Other intervention forms comprise transition programs for new nurses, assertiveness training, and education to improve workplace bullying knowledge and perception [16,17].

Recent studies involving Korean nurses reported that the workplace cyberbullying incidence rate was 8–10% [1,12]. Nevertheless, unlike in other occupational groups, the status survey on workplace cyberbullying in the nursing context is extremely limited [1]. Moreover, recent interventional studies on workplace bullying have focused mostly on new and general nurses [13,17,18]. Thus, there is a need to develop intervention programs for head nurses who can serve as agents for policy implementation at the organizational level, directly managing the nursing staff to ensure they follow all procedures and medical best practices and contributing significantly to building organizational culture [19].

Nurses experience time and space constraints due to the nature of their work. In nursing education, mobile and internet technology that enables continued learning via text, audio, video, and image data without time and space constraints has been expanding [20]. During the COVID-19 pandemic, effective training methods without direct contact became necessary. With the increase in non–face-to-face communication and mobile device usage [1], traditional training methods have been actively changing to mobile learning (m-learning) [21]. However, studies regarding traditional bullying and workplace cyberbullying have rarely assessed the effects of educational programs developed for nurses, which is a new paradigm [1,19,22]. Therefore, given the efficiency of cognitive rehearsals in traditional bullying training for head nurses, this study aimed to develop a version of the cognitive rehearsal program that is suitable for cyberbullying. The study applies the m-learning method to evaluate the effectiveness of the program and its effects on symptom experience, knowledge, perception, and turnover intention of head nurses.

**H1:** 
*Compared to the control group, the experimental group will experience fewer negative symptoms and exhibit a lower turnover intention.*


**H2:** 
*Compared to the control group, the experimental group will exhibit higher levels of perception and knowledge.*


## 2. Methods

### 2.1. Study Design

This study was evaluated with a nonequivalent control group pretest-posttest and a quasi-experimental design.

### 2.2. Setting and Participants

The study setting was a university hospital with 1700 beds located in “I” city (Republic of Korea). The hospital employed 78 female head nurses. The inclusion criteria were: worked as a head nurse for at least six months at the hospital, had no experience in workplace cyberbullying training, had a minimum of 10 workers within the department, and could attend workshops. Those whose job title was changed to general nurse were excluded. The total sample size for the effect evaluation was calculated using G-power 3.1.9.2. Moreover, the effect size from a previous study that examined the influence of a face-to-face cognitive rehearsal program for nurses [23] was used (d = 0.66). Calculations based on a *t*-test effect size of d = 0.66, significance level (α) of 0.05, and statistical power (1 − β) of 0.80 indicated that the minimum sample size was 30. Considering drop-outs, 35 participants were selected for the experimental and control groups.

The initial 70 candidates who volunteered to partake were recruited to participate in the pretest-posttest survey and the program. After excluding one control group participant due to incomplete responses, the data from the remaining 69 were included in the final analysis. For group allocation, cards with the letter “A” or “B” (experimental and control groups, respectively) were placed in a non-transparent box, from which each participant personally picked a card (Figure 1).

### 2.3. Procedures

A research assistant informed the participants about the study objectives during a one-hour, one-day head nurse workshop on 29 July 2021. The pretest was administered to the 70 participants to investigate their symptom experience, perception and knowledge of cyberbullying, and turnover intention.

The experimental group (*n* = 35) remained in the workshop venue to observe the workplace cyberbullying cognitive rehearsal m-learning program; when necessary, they were repeatedly directed to a password-protected Vimeo site throughout the day to watch the video using mobile phones or computers. The participants completed the posttest within 24 h after finishing the program and submitted it to the nursing department’s administrative office.

The control group (*n* = 35) was prohibited from watching the Vimeo video and did not receive any intervention. The following day, they completed the post-test questionnaire and submitted it to the nursing department’s administrative office. Those who wanted to complete the program were provided with the link to the Vimeo site and its password two days after the questionnaire’s completion. The participants who finished all questionnaires were provided a gift certificate (KRW 30,000).

### 2.4. Intervention

The program was developed between 1 April and 28 July 2021 using the Analysis, Design, Development, Implementation, and Evaluation (ADDIE) step process. ADDIE was proposed as evidence of instructional design by Peterson [24] and is widely employed for m-learning development procedures.

The cognitive rehearsal training program previously developed by Griffin [13] and Stagg et al. [14] was utilized to analyze the intervention. To examine participants’ needs, interviews were conducted individually with three nurses who had experience of resignation from work due to bullying. The major needs identified were methods for coping with workplace cyberbullying, followed by case definition and causes. For program content, previous studies [13,14,15,18,25,26] and individual interviews were used to identify the common contents. Accordingly, the following list of themes was established: theoretical concepts of workplace bullying; common bullying behaviors; consequences of bullying; what can be done about workplace cyberbullying; a total of 10 cognitive rehearsal cyberbullying scenarios; and a summary. The contents corresponding to each theme were reviewed (Table 1).

The program’s goals were established based on the analysis results and were designed to: (a) present a problem by exposing the participants to two cognitive rehearsal scenarios, (b) conduct the training program, and (c) respond after presenting 10 cognitive rehearsal scenarios again.

A professor with three years of research experience in workplace cyberbullying administered the 40-min training program to the nurses via video. An explanation of the program goals and learning methods was displayed on the screen, and a voice speech described the cases. After the nurses learned all the content, the final evaluation was performed.

The developed program’s content validity was assessed by a four-member expert panel, after which it was partially revised and supplemented. The content validity coefficient ranged from 0.95 to 1.00. Regarding the revised and supplemented content, a national-level response was added to the response methods. The completed program was uploaded to Vimeo for playback using a mobile phone or a computer. The use of Vimeo software required a set password to access any content that was copyright patented (Figure 2).

### 2.5. Measurements

#### 2.5.1. Symptom Experience

The nurses’ symptom experience was measured using the brief symptom tool, originally developed as the Brief Symptom Inventory-18 by Derogatis [27] and subsequently validated as a Korean scale by Park et al. [28]. It consists of eighteen items divided into four categories: five, six, four, and three items for somatization, depression, anxiety, and phobic anxiety, respectively. Experience of each symptom in the past seven days was rated on a 5-point Likert scale (1 = “not at all” to 5 = “extremely severe”); higher total scores indicated greater experience of negative symptoms. The scale’s reliability calculated using Cronbach’s alpha was 0.89 in Park et al.’s study [28] and 0.91 in the present research.

#### 2.5.2. Knowledge of Workplace Cyberbullying

The measure for knowledge of workplace cyberbullying was developed by the researcher by referring to the knowledge assessment of bullying among medical students in Quetta [29]. The content validity index (CVI) was tested by two head nurses, one chief nurse, and one nursing professor. The final 19 items were selected based on a pilot study of five nurses. They comprised cyberbullying’s definition (two items), causes (two items), influence (two items), subjects at risk and response (three items), and the judgment of cases (10 items: “received rude work demands”, “received rude work messages”, “criticized unfairly”, “received rude work messages expressed in offensive language”, “ignoring my opinions in intergroup communication”, and “receiving unfair personal criticism”, “hearing unfavorable rumors or chitchat about me”, “sharing personal information without my consent”, “unjustly doubting my abilities”, “removing myself from group chats with my online coworkers”).

Each correct and incorrect or missing answer was assigned one and zero points, respectively. The total score was converted to a 100-point scale; higher scores indicated greater levels of knowledge. The final expert validity was a CVI of 0.95. The pilot and the present study’s reliabilities using the Kuder–Richardson formula 20 were 0.65 and 0.72, respectively.

#### 2.5.3. Perception of Workplace Cyberbullying

The researcher constructed the tool for the perception of workplace cyberbullying by referring to the scale employed to measure the perception of bullying among medical students in Quetta [29]. It consisted of six items (“the workplace bullying’s definition”, “the workplace bullying’s causes”, “the workplace bullying’s negative influence”, “I believe workplace bullying is illegal”, “the workplace bullying cases”, and “the workplace bullying’s coping methods”); the CVI was assessed by four experts. Each item was rated on a 5-point Likert scale (1 = “not at all” to 5 = “completely agree”); higher total scores indicated greater levels of perception. The scale’s reliability, computed using Cronbach’s alpha, was 0.89 and 0.91 in the pilot and the present study, respectively.

#### 2.5.4. Turnover Intention

Turnover intention refers to the purpose of voluntarily leaving the present job for another institution [30]. It was measured using a tool originally employed by Mobley et al. [30] and translated into Korean by Shin and Cho [31]. It comprises five items, each rated on a 5-point Likert scale (1 = “not at all” to 5 = “completely agree”); higher total scores signified greater turnover intentions. In this study, its reliability, calculated using Cronbach’s alpha, was 0.84.

### 2.6. Data Analysis

SPSS/WIN 23.0 (IBM Corp, Armonk, NY, USA) was used for data analysis. The homogeneity of the characteristics and pretest-dependent variables between the experimental and control groups were examined by the frequency, mean, chi-square tests, Fisher exact tests, and *t*-tests. The pretest-posttest differences’ normality in the dependent variables of the two groups was examined using the Shapiro–Wilk test. According to the aforementioned normality, an independent *t*-test (perception) or the Mann–Whitney U test (symptom experience, knowledge, and turnover intention) was utilized to analyze the differences between the two groups. Furthermore, the statistical significance was set at *p* < 0.05. For the measurement tools’ reliability, the Kuder–Richardson formula 20 and Cronbach’s alpha were used.

### 2.7. Ethical Considerations

This study was approved by the Institutional Review Board of “G” University (IRB NO.1044396-202106-HR-129-01) and adhered to the Declaration of Helsinki. The participants were informed that the collected data would be anonymously codified, would only be used for this study’s purposes, and that they could refuse or withdraw their consent at any time. Additionally, written informed consent was obtained from each participant before the study commenced.

## 3. Results

### 3.1. Participants’ General Characteristics

All participants (*n* = 69) were female and did not work in a three-shift system. The clinical experience’s mean length was 28.24 ± 5.10 and 26.79 ± 4.59 years in the experimental and control groups, respectively. Regarding the experience of face-to-face workplace bullying in the past six months, two cases each in the experimental (5.7%) and control (5.9%) groups were reported. With respect to facing workplace cyberbullying in the past six months, there were three (8.6%) and no (0%) cases in the former and latter, respectively. The pre-intervention pretest results demonstrated no statistically significant differences in the general characteristics and dependent variables between the two groups (Table 2).

### 3.2. Differences in Symptom Experience, Knowledge and Perception of Workplace Cyberbullying, and Turnover Intention between the Experimental and Control Groups

There was an increase in the experimental group’s scores from the pretest to the posttest for symptom experience (3.10 ± 1.12 to 3.47 ± 1.13 points), knowledge of workplace cyberbullying (84.06 ± 11.59 to 95.19 ± 5.32 points), and perception of workplace cyberbullying (3.98 ± 0.52 to 4.45 ± 0.82 points). Consistent with H2, a statistically significant difference was observed between the experimental and control groups’ symptom experience (*U* = 414.50, *p* = 0.030), knowledge of workplace cyberbullying (*U* = 319.00, *p* = 0.001), and perception of workplace cyberbullying (*t* = 2.01, *p* = 0.048).

The turnover intention in the experimental group decreased from 2.04 ± 0.83 to 1.94 ± 0.80 points at the pretest and posttest, respectively. However, there was no statistically significant difference between both groups (*U* = 538.50, *p* = 0.485).

As a result of hypothesis verification, the difference values of the experimental group compared to the control group were confirmed to be significant for symptom experiences, knowledge, and perception. However, there was no statistically significant difference between both groups for turnover intention; H1 is not supported (Table 3).

## 4. Discussion

Workplace cyberbullying in nursing organizations is a new concept whose influences should be evaluated by developing effective intervention programs on the same level as face-to-face bullying. However, such programs have thus far been unavailable for head nurses, who serve as organizational managers [19]. This study’s novelty is that it developed and evaluated the efficiency of a cyberbullying cognitive rehearsal program for head nurses using m-learning.

Previous studies that directly applied cognitive rehearsal interventions for workplace bullying for nurses [13,14,15] used a quasi-experimental design. They reported that two hours of intervention yielded positive effects; the intervention increased knowledge and awareness and decreased bullying prevalence. However, these studies had a small sample size of only 10–26 nurses, with no control group. Thus, to overcome these shortcomings, this research employed an experimental control design based on a sample size calculation. Further, a randomized controlled trial by Kang et al. [23] utilized a 20-h, 10-session program that was applied for a relatively long 5-week period; it did not affect bullying prevalence and symptom experience. Despite the prolonged time and effort, the educational impacts increased interpersonal relationships and decreased turnover intention; thus, there is a need to consider newer instructive methods, such as m-learning, which are additionally cost-effective.

Different from existing studies, the cognitive rehearsal program for cyberbullying applied in this study was proven effective as a new educational technique using mobile technology [13,14,15,23]. Regarding the cyberbullying themes for head nurses, due to insufficient existing interventional studies, this study developed 10 rehearsals (five work- and person-related cyberbullying scenarios each), which could be done in cyberspace as opposed to the standardized face-to-face rehearsals. While they were tested for expert validity, an additional examination is needed for their universal use as standardized scenarios. Furthermore, the development of suitable scenarios for each country is required.

This research’s effect evaluation of the two groups demonstrated statistically significant differences in symptom experience, knowledge, and perception of workplace cyberbullying, but not in turnover intention. Similar to a face-to-face workplace bullying intervention study that applied cognitive rehearsal, the experimental group’s knowledge and perception scores in this study were significantly higher than that of the control group, thus indicating the effectiveness of cognitive rehearsal m-learning programs for workplace cyberbullying [14,15]. In Kang et al. [23], the effects of face-to-face bullying intervention on staff and charge nurses reported no effect on symptom experience. However, the present study showed a significant increase in head nurses’ symptom experience. Head nurses hold a leadership position and become agents of policy implementation at the organizational level. They serve as role models for other nurses and managers and contribute to building the organization’s culture [6,19]. Therefore, it is believed that, while head nurses serve as the mediators and managers, as well as the link between individual team members and the organization, their psychologically and physically negative symptoms increase due to new concepts and education training about cyberbullying [2,10]. In previous studies, cyberbullying experience elevated negative physical symptoms [12]; thus, it is supposed that the indirect experience with 10 scenarios for cyberbullying cognitive rehearsal presented difficulties and distress for head nurses who had to manage their team members. Therefore, it is recommended that the subjects for symptom experience be changed to nurses, or qualitative research on head nurses be conducted along with replication studies when evaluating the effects of intervention programs. In addition, this study immediately evaluated the intervention effect after only 24 h, increasing the head nurses’ symptom experience. A longer follow-up period may help identify whether symptom experience decreased over time after acute exposure to the intervention in the short term.

Unlike previous studies, this study’s program did not affect turnover intention [18,23]. It is assumed that this was because the program had a relatively short video running time of 40 min and had only 24-h access, which was different from prior studies that had an operating time of 2–20 h [13,14,15,23]. Moreover, contrary to previous research that conducted posttests on turnover intention after four or eight weeks, the present study conducted it 24 h after applying the program; thus, the evaluation of middle-to-long term effects is needed in future studies [18,23]. The program can also be applied with objective structured clinical examinations (OSCEs), which are frequently included in the curriculum of nursing schools. An earlier investigation [32] that assessed the efficiency of OSCEs for clinical nurse specialists revealed their beneficial educational effects for coaching competencies for behavior change. Similarly, new dependent variables must be taken into account when determining different ways to measure the educational impact of scenario-based mobile programs. Nursing students may benefit from an efficient educational program if the cyberbullying scenario from the actual clinical setting and the current scenario created by OSCEs are combined [32,33].

This study had several limitations. First, the participants only included head nurses; thus, repeating the effect evaluation with general and chief nurses is suggested. In addition, all participants in the study were female, making it difficult to compare the effects of interventions based on sex. To evaluate the effects of sex on such interventions, a future study that includes male subjects is suggested. Second, randomization was not possible because the experimental and control groups were selected out of convenience. Subsequent studies should consider randomized sampling. Third, due to insufficient research on cyberbullying, the program operation and application durations were set to a relatively short time. Therefore, additional tests on the effective operational time and method are needed. Regarding the effect evaluation’s dependent variables, symptom experience requires more replication studies with an expanded research population and comparison with general staff nurses. Moreover, other dependent variables, such as the cyberbullying reporting rate, should be considered. Lastly, turnover intention remained unaffected when it was evaluated immediately after the program’s application; thus, it is essential to assess the effect by extending the evaluation period.

## 5. Conclusions

This study’s cognitive rehearsal m-learning program for workplace cyberbullying provided indirect experience of cyberbullying cases that are difficult to learn directly in clinical settings. The results showed that the developed program had a negative effect and increased head nurses’ physical symptoms. However, it also had a significantly positive educational impact by enhancing their knowledge and perception of workplace cyberbullying. Thus, the developed program could be an effective educational strategy in the nursing field.

### Relevance for Clinical Practice

The application of a cognitive rehearsal m-learning program can be used to effectively train head nurses on workplace cyberbullying within a short period. It can efficiently enhance their knowledge and perceptions about cyberbullying. Therefore, such a program could enable head nurses to more successfully manage their staff. Additionally, the program could also be expanded to various target populations, nursing education, and medical institutions.

## Figures and Tables

**Figure 1 healthcare-11-02041-f001:**
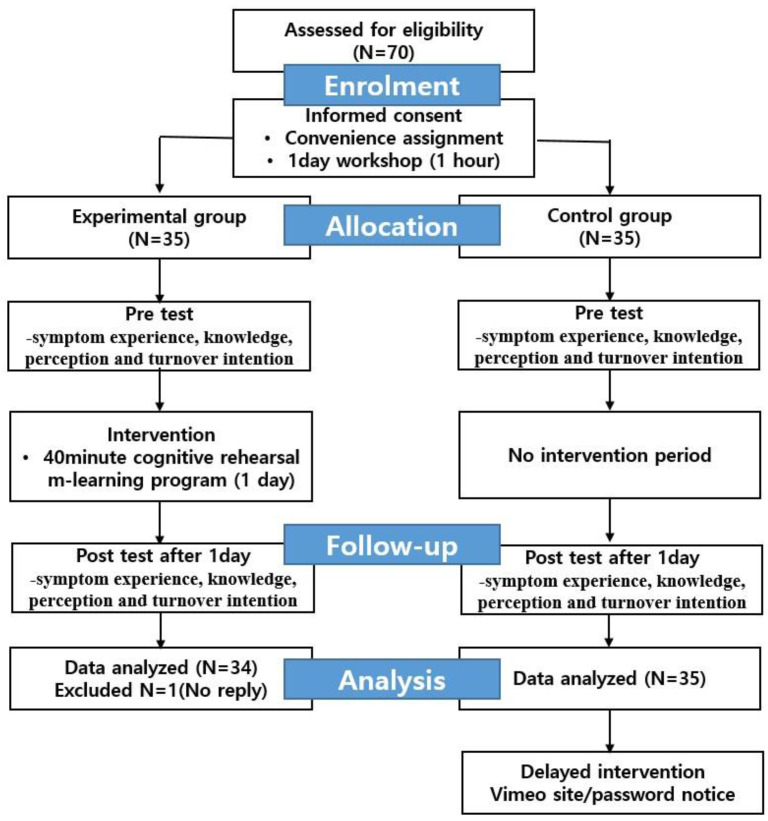
Study design.

**Figure 2 healthcare-11-02041-f002:**
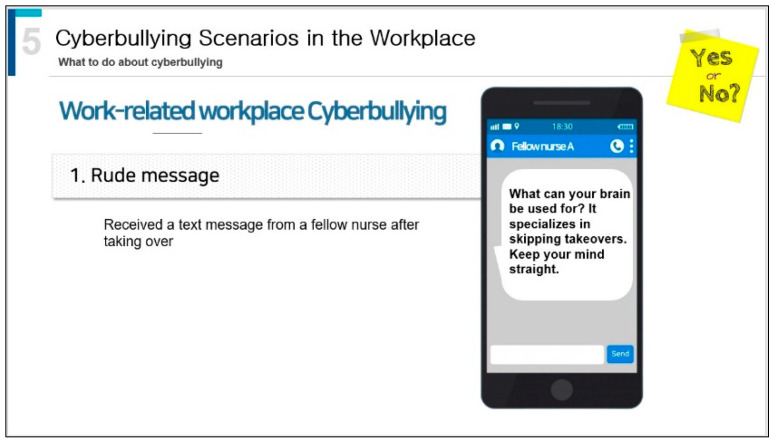
Sample page of the developed mobile learning program.

**Table 1 healthcare-11-02041-t001:** Content Validity of the workplace cyberbullying cognitive rehearsal m-learning program.

Categories	Construct	Item	Content Validity
1. Theoretical concepts of workplace cyberbullying.	Text, picture, photograph, voice recording, and video lecture	General information	1.00
		Question: Is the following situation workplace cyberbullying? (1) Work-related cyberbullying scenario 1 (2) Person-related cyberbullying scenario 1	
		Definition; other terms	
		Common theories regarding workplace cyberbullying	
		Prevalence of workplace bullying	
		Target characteristics	
		Causes and effects of workplace cyberbullying	
2. Common workplace cyberbullying behaviors.	Text, picture, photograph, and voice recording	Characteristics of cyberbullying	1.00
		Types of workplace cyberbullying	
		(1) 10 Work-related cyberbullying behaviors (2) 11 Person-related cyberbullying behaviors	
3. What can be done about workplace cyberbullying?	Text, picture, photograph, and voice recording	Individual; micro level (1) 11 ways to handle workplace cyberbullying	0.95
		Organizational; meso level	
		Industry and national; macro level (1) Legal response (2) School Violence Prevention Law (3) Workplace Harassment Prevention Law	
4. Workplace cyberbullying 10 cognitive rehearsal scenarios.	Text, picture, photograph, voice recording, and video lecture	Demonstration scenarios Practice scenarios	1.00
		(1) 5 Work-related cyberbullying scenarios (2) 5 Person-related cyberbullying scenarios	
5. Summary.	Text, picture, photograph, voice recording, and video lecture	Summary	1.00
Total			0.97

**Table 2 healthcare-11-02041-t002:** Homogeneity tests between the experimental and control groups (*n* = 69).

Variables (Range)	Categories	Experimental (*n* = 35) *n* (%) or Mean ± SD	Control (*n* = 34) *n* (%) or Mean ± SD	χ^2^ or *t*	*p*
Gender	Female	35(100.0)	34(100.0)		
Age (in years)		50.60 ± 5.07	48.76 ± 4.81	1.541	0.128
Marital status	Married	29(82.9)	28(82.4)	0.003	0.998
	Unmarried	5(14.3)	5(14.7)		
	Others	1(2.9)	1(2.9)		
Education	Associate degree	1(2.9)	0(0.0)	1.348	0.718
	Bachelor	8(22.9)	6(17.6)		
	Master	25(71.4)	27(79.4)		
	PhD	1(2.9)	1(2.9)		
Department	General ward	17(48.6)	13(38.2)	4.274	0.370
	Intensive care unit	3(8.6)	5(14.7)		
	Operation room	2(5.7)	6(17.6)		
	Outpatient unit	4(11.4)	5(14.7)		
	Others	9(25.7)	5(14.7)		
Position	Head nurse	35(100.0)	34(100.0)		
Salary	<5000	1(2.9)	2(5.9)	1.410	0.703
(10,000 KRW/year)	5000–<7000	27(77.1)	27(79.4)		
	7000–<9000	6(17.1)	5(14.7)		
	≥9000	1(2.9)	0(0.0)		
Clinical experience length (in years)		28.24 ± 5.10	26.79 ± 4.59	0.926	0.339
Three-shift system	No	35(100.0)	34(100.0)		
Experience with face-to–face workplace bullying	Yes	2(5.7)	2(5.9)	0.001 ^a^	1.000
(in the past 6 months)	No	33(94.3)	32(94.1)		
Experience with workplace cyberbullying	Yes	3(8.6)	0(0.0)	3.047 ^a^	0.239
(in the past 6 months)	No	32(91.4)	34(100.0)		
Symptom experience (1–5)		3.10 ± 1.12	2.89 ± 1.04	0.809	0.422
Knowledge of workplace cyberbullying (0–100)		84.06 ± 11.59	88.08 ± 8.22	−1.657	0.102
Perception of workplace cyberbullying (1–5)		3.98 ± 0.52	3.76 ± 0.65	1.575	0.120
Turnover intention (1–5)		2.04 ± 0.83	2.29 ± 0.66	−1.372	0.175

Abbreviations: SD, standard deviation. ^a^ Fisher exact test.

**Table 3 healthcare-11-02041-t003:** Differences in the values for symptom experience, knowledge and perception of workplace cyberbullying, and turnover intention between the experimental and control groups (*n* = 69).

Variables	Group	Pretest	Posttest	Difference	*t* or *U*	*p*
		M ± SD	M ± SD	M ± SD		
Symptom experience	Exp.(*n* = 35)	3.10 ± 1.12	3.47 ± 1.13	0.37 ± 0.76	414.50 *	0.030
	Con.(*n* = 34)	2.89 ± 1.04	2.86 ± 1.10	−0.03 ± 0.50		
Knowledge of workplace cyberbullying	Exp.(*n* = 35)	84.06 ± 11.59	95.19 ± 5.32	11.13 ± 11.89	319.00 *	0.001
	Con.(*n* = 34)	88.08 ± 8.22	88.85 ± 9.50	0.77 ± 11.15		
Perception of workplace cyberbullying	Exp.(*n* = 35)	3.98 ± 0.52	4.45 ± 0.82	0.47 ± 0.99	2.01	0.048
	Con.(*n* = 34)	3.76 ± 0.65	3.83 ± 0.65	0.07 ± 0.60		
Turnover intention	Exp.(*n* = 35)	2.04 ± 0.83	1.94 ± 0.80	−0.09 ± 0.58	538.50 *	0.485
	Con.(*n* = 34)	2.29 ± 0.66	2.25 ± 0.77	−0.03 ± 0.40		

Abbreviations: Exp. = experimental group; Con. = Control group; M = mean; SD = standard deviation; *t* = independent *t*-test; *U* = Mann–Whitney *U* test *.

## Data Availability

The data can be requested from the corresponding author for proper reasons.

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
