# Peer review of "A Pilot Study to Examine the Effects of a Workplace Cyberbullying Cognitive Rehearsal Mobile Learning Program for Head Nurses: A Quasi-Experimental Study"

_healthcare, 2023, doi:10.3390/healthcare11142041_

Round 1

Reviewer 1 Report

The design and methodology are sound, adhering to the appropriate tests for the comparisons made.

The topic is novel in terms of the type of intervention, although perhaps the subject matter implies in advance a higher risk of exposure to cyberbullying than is apparent from the review of studies. Nevertheless, it is still a problem of social interest.

Although I have no major objections, I would like to point out some details for improvement:

- The section on Knowledge of workplace cyberbullying needs to be better clarified.

- Add some information on the effect size on the dependent variables presented.

- In the section on limitations of the study, reference should be made to the fact that it is a sample of volunteers and how this may affect external validity.

Author Response

- The section on Knowledge of workplace cyberbullying needs to be better clarified.

  • We specified the 10 cases that should be highlighted in terms of knowledge of workplace cyberbullying in lines 192-197: “received rude work demands,” “received rude work messages,” “criticized unfairly,” “received work messages expressed in offensive language,” “ignoring my opinions in intergroup communication,” and “receiving unfair personal criticism,” “hearing unfavorable rumors or chitchat about me,” “sharing personal information without my consent,” “unjustly doubting my abilities,” “removing myself from group chats with my online coworkers.”

- Add some information on the effect size on the dependent variables presented.

  • We added lines 107-109 to clarify that the effect size was determined based on a previous research by Kang et al. (2017): “The total sample size for the effect evaluation was calculated using G-power 3.1.9.2. Moreover, the effect size from a previous study that examined the influence of a face-to-face cognitive rehearsal program for nurses was used (d=0.66).”

- In the section on limitations of the study, reference should be made to the fact that it is a sample of volunteers and how this may affect external validity.

  • In accordance with the suggestions of other reviewers, we clarified in lines 343-345 that the study participants were volunteers. They were “convenience samples” rather than random selections. As a result, the entire research procedure was changed to a "quasi-experimental study design.” Accordingly, we listed the study’s lack of randomization as a limitation: “Second, randomization was not possible because the experimental and control groups were selected out of convenience. Subsequent studies should consider randomized sampling.”

Reviewer 2 Report

Dear Editor and Authors. The article needs a strengthening of the theoretical framework, using meta-analysis, systematic reviews and longitudinal studies to further address cyberbullying and its moderating variables. Also, the study lacks hypotheses, add them and explain in the results whether they are confirmed or refuted. Likewise, the references are very scarce, and I believe that this is why the theoretical framework is a bit thin. The methodology is very well designed even though the sample is very, very small. The study has limitations but does not talk about foresight and practical applications, please add them.

Author Response

Response to Reviewer 2 Comments

The article needs a strengthening of the theoretical framework, using meta-analysis, systematic reviews and longitudinal studies to further address cyberbullying and its moderating variables.

  • We added the lines below to strengthen the theoretical framework. As suggested, a meta-analysis, systematic reviews, and longitudinal studies were used to further address the difference between workplace cyberbullying and cyberbullying and its moderating variables:
  • Lines 36-39: “While cyberbullying has been extensively studied in the literature on childhood adolescence and emerging adulthood, only recently has systematic research begun to focus on cyberbullying at the workplace.”
  • Lines 40-46: “Based on a literature review and meta-analysis, a number of theories serve as the theoretical foundation for the causes and effects of workplace cyberbullying. The Affective Events Theory (AET) describes how the experience of emotions like anger or fear can cause affective work events to result in different work attitudes and affective driven behaviors. Initially intended as a theory of work satisfaction, AET integrated prior knowledge on emotions to provide future directions for the research on emotions in the organizational context.”

  • Lines 47-59: “Cyberbullying, which occurs in cyberspace, is more serious than traditional bullying, as victims can be continuously tormented regardless of their physical location. Cyberbullying can be identified based on a socio-ecological model, which has previously also been used to examine traditional workplace bullying in nursing contexts. Although not a complete list of contributing factors, the socio-ecological model may serve as a starting point for enhancing our understanding of how workplace cyberbullying occurs. Additionally, its contributing factors may serve as a springboard for management, intervention, and prevention initiatives against workplace cyberbullying. Based on the socio-ecological model, workplace cyberbullying occurs in three levels: individual (micro), organizational (meso), and industry and national (macro); its outcomes have several influences on each level. At the individual level, workplace cyberbullying negatively impacts the victims’ psychological and physical well-being. At the organizational level, according to the AET theory, it has an effect on turnover intention to leave an organization and job satisfaction.”

Also, the study lacks hypotheses, add them and explain in the results whether they are confirmed or refuted.

  • We have more explicitly stated our research hypotheses in lines 91-95 (Section 2.2): (1) “In comparison to the control group, the difference of experimental group will experience fewer negative symptom experiences and have a lower turnover intentio” (2) “In comparison to the control group, the difference of experimental group will have higher levels of perception and knowledge.”
  • We further explained in lines 264-267 whether the hypotheses were supported or not: “As a result of hypothesis verification, the difference values of the experimental group compared to the control group were confirmed to be significant for symptom experiences, knowledge, and perception. However, there was no statistically significant difference between both groups for turnover intention; H1 is not supported (Table 3).”

Likewise, the references are very scarce, and I believe that this is why the theoretical framework is a bit thin.

  • We have added the following references, and enriched the theoretical framework in the introduction (lines 36-59):

Smith, P.K.; Mahdavi, J.; Carvalho, M.; Fisher, S.; Russell, S.; Tippett, N. Cyberbullying: Its nature and impact in secondary school pupils. J Child Psychol Psychiatry, 2008, 49, 376e385. DOI:10.1111/j.1469-7610.2007.01846.x.

Coyne, I.; Farley, S.; Axtell, C.; Sprigg, C.; Best, L.; Kwok, O. Understanding the relationship between experiencing workplace cyberbullying, employee mental strain and job satisfaction: A dysempowerment approach. Int J Hum Resour Manag. 2017, 28, 945–972. DOI:10.1080/09585192.2015.1116454.

Choi, J.; Park, M. Effects of nursing organisational culture on face-to-face bullying and cyberbullying in the workplace. J Clin Nurs. 2019, 28, 2577–2588. DOI:10.1111/jocn.14843.

Forssell, R.C. Cyberbullying in a boundary blurred working life: distortion of the private and professional face on social media. Qual Res Organ Manag. 2019. 15, DOI:89-107.10.1108/QROM-05-2018-1636.

Vranjes, I.; Baillien, E.; Vandebosch, H.; Erreygers, S.; Witte, H.D. The dark side of working online: Towards a definition and an Emotion Reaction model of workplace cyberbullying. Comput Hum Behav. 2017, 69, 324–334. DOI:10.1016/j.chb.2016.12.055.

Weiss, H.M.; Cropanzano, R. Affective events theory: A theoretical discussion of the structure, causes and consequences of affective experiences at work. In Research in Organizational Behavior; Staw B.M., Cummings, L.L., Eds.; JAI Press: Greenwich, CT, USA, 1996; p. 1e74.

Kowalski, R.M.; Giumetti, G.W.; Schroeder, A.N.; Lattanner, M.R. Bullying in the digital age: A critical review and meta-analysis of cyberbullying research among youth. Psychol Bull. 2014, 140, 1073­–1137. DOI: 10.1037/a0035618.

Cuevas, H.E.; Timmerman, G.M. Use of an objective structured clinical examination in clinical nurse specialist education. Clin Nurse Spec. 2016, 30, 172–176. DOI: 10.1097/NUR.0000000000000201.

Smith, C.R.; Gillespie, G.L.; Brown, K.C.; Grubb, P.L. Seeing students squirm: Nursing students’ experiences of bullying behaviors during clinical rotations. J Nurs Educ. 2016, 55, 505–513. DOI: 10.3928/01484834-20160816-04.

The methodology is very well designed even though the sample is very, very small. The study has limitations but does not talk about foresight and practical applications, please add them.

  • We describe in the lines 107-109 how the number of samples was determined using a statistical program based on previous research: “The total sample size for the effect evaluation was calculated using G-power 3.1.9.2. Moreover, the effect size from a previous study that examined the influence of a face-to-face cognitive rehearsal program for nurses was used (d=0.66). Calculations based on a t-test effect size of d=0.66, significance level (α) of .05, and statistical power (1 − β) of .80 indicated that the minimum sample size was 30. Considering drop-outs, 35 participants each were selected for the experimental and control groups.”
  • Following your recommendation, we also discuss in the following lines the practical implications and foresight:
  • Section 5.1 (lines 363-369):The application of a cognitive rehearsal m-learning program can be used to effectively train head nurses on workplace cyberbullying within a short period. It can efficiently enhance their knowledge and perceptions about cyberbullying. Therefore, such a program could enable head nurses to more successfully manage their staff. Additionally, the program could also be expanded to various target populations, nursing education, and medical institutions.”
  • Lines 330-338: “The program can also be used with objective structured clinical examinations (OSCEs), which are frequently used in the curriculum of nursing schools. An earlier investigation that assessed the efficiency of OSCEs for clinical nurse specialists revealed beneficial educational effects for coaching competencies for behavior change. Similar to this, new dependent variables must be taken into account when determining different ways to measure the educational impact of scenario-based mobile programs. Nursing students may benefit from an efficient educational program if the cyberbullying scenario from the actual clinical setting and the current scenario created by OSCEs are combined.”

Reviewer 3 Report

Dear authors,

Thank you for let me reviewing your manuscript titled "The effects of a workplace cyberbullying cognitive rehearsal mobile learning program for head nurses: A quasi-experimental pragmatic study"

In my opinion is an interesting manuscript, linked to the real day to day work, about a relevant subject, and therefore it deserves to be published.

Let me please, to propose some questions and considerations in order to improve your manuscript.

Abstract,

I miss mentioning the statistic employed in the methodology abstract´s section and the use of validated scales to measure the variables. Line 20-21 "quasi-randomized". I wonder why? It looks like a convenience sample. How was the randomized method?. Line 24 "effective program" Why was effective? Lines 26-27 "Nursing organizations´head nurses need to have high knowledge and positive perception of cyberbullying" is a result or a conclusion of your study? Why?

In Introduction, lines 38-39 "Bullying in nursing organizations is more common than in other establishments", please support this statement with reference/s.

Methods

Line 90, "...candidates who volunteered to partake.." These words are an evidence about that authors employed a "convenience sample", not a random or "quasi-randomized" one. How did authors control bias in their study?

Line 146, "...uploaded to Vimeo for playback using a mobile phone or a computer..." I miss a paragraph about the software license or patent, or the permissions required for its use.

Discussion

I suggest to introduce a paragraph in discusión about the utility of your study to the nursing students training and evaluation, in connection with the sentences in lines 54-55 of the introduction. For example to the Objetive Structured Clinical Examination, working with "cyberbullying scenarios". You have one reference, as an example about OSCE in:                                   Cuevas Heather E, Timmerman GM. Use of an Objective Structured Clinical Examination in Clinical Nurse Specialist Education. Clinical Nurse Specialist 2016; 30 (3):172-176. Doi: 10.1097/NUR.0000000000000201.

And perhaps, comparing it with some "virtual reality experiences".

In limitations I see the fact that it is a not random sample as a limitation.

Conclusions

Is the first sentence (lines 323-324) a conclusion of authors manuscript? Pleas consider to re write it or eliminate it.

References, please review it and notice if all the references with the same authors manuscript are essential, in order to avoid an undesirable "self citations".

Congratulations!

Author Response

Dear Reviewers, thank you for your valuable comments. We highly appreciate your input in helping us improve our paper. We have taken into consideration all of your feedback, and present our responses to your comments below.  

Response to Reviewer 3 Comments

Abstract,

I miss mentioning the statistic employed in the methodology abstract´s section and the use of validated scales to measure the variables.

  • We have indicated the statistical methods employed in the abstract in lines 18-21: “Variables with proven reliability were used in the program effect measurement. The differences between the experimental and control groups were examined using an independent t-test (perception) or the Mann–Whitney U test (symptom experience, knowledge, and turnover intention).”

Line 20-21 "quasi-randomized". I wonder why? It looks like a convenience sample. How was the randomized method?.

  • We have revised the statement to clarify that the study is a quasi-experimental design in line 17: “This study was evaluated using a nonequivalent control group pretest-posttest and a quasi-experimental”

 Line 24 "effective program" Why was effective? Lines 26-27 "Nursing organizations´head nurses need to have high knowledge and positive perception of cyberbullying" is a result or a conclusion of your study? Why?

  • We added lines 23-27 to indicate that the developed program could be effective when applied to the nursing field. We also emphasized how the role of head nurses requires them to have knowledge about cyberbullying in order to recognize and prevent it at the workplace: “This program could be applied as a valuable educational strategy in the nursing field. Head nurses act as intermediaries between individuals and the organization. Therefore, they must respond with in-depth knowledge and perceptions of cyberbullying to fulfill their responsibilities of identifying, mediating, and managing cyberbullying among hospital team members.”

In Introduction, lines 38-39 "Bullying in nursing organizations is more common than in other establishments", please support this statement with reference/s.

  • We cited previous research by Kim and Choi (2021) and Natalia et al. (2017) in line 32 to support the above statement.

Methods

Line 90, "...candidates who volunteered to partake." These words are an evidence about that authors employed a "convenience sample", not a random or "quasi-randomized" one. How did authors control bias in their study?

  • Following your recommendation, we have changed all instances of “quasi-randomized” in the entire document to “quasi-experimental study design.”

Line 146, "...uploaded to Vimeo for playback using a mobile phone or a computer..." I miss a paragraph about the software license or patent, or the permissions required for its use.

  • We have added lines 169-170 to indicate how a password was required to access the content: “The use of Vimeo software required a set password in order to access any content that was copyright patented.”

Discussion

I suggest to introduce a paragraph in discusión about the utility of your study to the nursing students training and evaluation, in connection with the sentences in lines 54-55 of the introduction. For example to the Objetive Structured Clinical Examination, working with "cyberbullying scenarios". You have one reference, as an example about OSCE in:  Cuevas Heather E, Timmerman GM. Use of an Objective Structured Clinical Examination in Clinical Nurse Specialist Education. Clinical Nurse Specialist 2016; 30 (3):172-176. Doi: 10.1097/NUR.0000000000000201. And perhaps, comparing it with some "virtual reality experiences".

  • We have added lines 330-338 to expound on the utility of our study to the training and evaluation of nursing students. We also cited the work of Smith et al. (2016) in line 336: “The program can also be applied with objective structured clinical examinations (OSCEs), which are frequently included in the curriculum of nursing schools. An earlier investigation that assessed the efficiency of OSCEs for clinical nurse specialists revealed their beneficial educational effects for coaching competencies for behavior change. Similarly, new dependent variables must be taken into account when determining different ways to measure the educational impact of scenario-based mobile programs. Nursing students may benefit from an efficient educational program if the cyberbullying scenario from the actual clinical setting and the current scenario created by OSCEs are combined.”

Cuevas, H.E.; Timmerman, G.M. Use of an objective structured clinical examination in clinical nurse specialist education. Clin Nurse Spec. 2016, 30, 172–176. DOI: 10.1097/NUR.0000000000000201.

Smith, C.R.; Gillespie, G.L.; Brown, K.C.; Grubb, P.L. Seeing students squirm: Nursing students’ experiences of bullying behaviors during clinical rotations. J Nurs Educ. 2016, 55, 505–513. DOI: 10.3928/01484834-20160816-04.

In limitations I see the fact that it is a not random sample as a limitation.

  • We added lines 343-344 to explain why randomization was not possible in the following statement: “Second, randomization was not possible because the experimental and control groups were selected out of convenience. Subsequent studies should consider randomized sampling.”

Conclusions

Is the first sentence (lines 323-324) a conclusion of authors manuscript? Pleas consider to re write it or eliminate it.

  • Following your suggestion, we have eliminated the first sentence in lines 323-324 of the original submission.

References, please review it and notice if all the references with the same authors manuscript are essential, in order to avoid an undesirable "self citations".

  • We have reviewed all references to ensure only those that are essential were included.

Congratulations!

Round 2

Reviewer 2 Report

Dear author and publisher,

The article has improved considerably. My compliments.

Author Response

The article has improved considerably. My compliments.

--> Thanks. 
